# Role of Nrf2 Nucleus Translocation in Beauvericin-Induced Cell Damage in Rat Hepatocytes

**DOI:** 10.3390/toxins14060367

**Published:** 2022-05-25

**Authors:** Jiabin Shi, Yaling Wang, Wenlin Xu, Guodong Cai, Hui Zou, Yan Yuan, Jianhong Gu, Zongping Liu, Jianchun Bian

**Affiliations:** 1College of Veterinary Medicine, Yangzhou University, 12 Wenhui East Road, Yangzhou 225009, China; mx120200948@yzu.edu.cn (J.S.); wyl11122022@163.com (Y.W.); xwl1009yl@163.com (W.X.); dx120180126@yzu.edu.cn (G.C.); zouhui@yzu.edu.cn (H.Z.); yuanyan@yzu.edu.cn (Y.Y.); jhgu@yzu.edu.cn (J.G.); liuzongping@yzu.edu.cn (Z.L.); 2Jiangsu Co-Innovation Center for Prevention and Control of Important Animal Infectious Diseases and Zoonoses, Yangzhou 225009, China; 3Joint International Research Laboratory of Agriculture and Agri-Product Safety of the Ministry of Education of China, Yangzhou University, Yangzhou 225009, China

**Keywords:** beauvericin, apoptosis, oxidative stress, Nrf2, autophagy

## Abstract

Beauvericin (BEA), a food-borne mycotoxin metabolite derived from the fungus *Beauveria Bassiana*, is proven to exhibit high hepatotoxicity. However, the molecular mechanism underlying BEA-induced liver damage is not fully understood. Herein, the effect of Nrf2 nuclear translocation-induced by BEA in hepatocytes was investigated. CCK8 solution was used to determine the appropriate concentrations of BEA (0, 1, 1.5 and 2 μmol/L), and BRL3A cells were then exposed to different concentrations of BEA for 12 h. Our results reveal that BEA exposure is associated with high cytotoxicity, lowered cell viability, damaged cellular morphology, and increased apoptosis rate. BEA could lead to oxidative damage through the overproduction of ROS and unbalanced redox, trigger the activation of Nrf2 signaling pathway and Nrf2 nuclear translocation for transcriptional activation of downstream antioxidative genes. Additionally, BEA treatment upregulated the expression of autophagy-related proteins (LC3, p62, Beclin1, and ATG5) indicating a correlation between Nrf2 activation and autophagy, which warrants further studies. Furthermore, ML385, an Nrf2 inhibitor, partially ameliorated BEA-induced cell injury while CDDO, an Nrf2 activator, aggravated liver damage. The present study emphasizes the role of Nrf2 nuclear translocation in BEA-induced oxidative stress associated with the hepatotoxic nature of BEA.

## 1. Introduction

Beauvericin (BEA), an emerging mycotoxin metabolite produced mainly by *Beauveria bassiana* and *Fusarium* species [1], is widely found in cereals or cereal-based products and is highly noxious to various cells in in vitro studies [2,3]. Interestingly, BEA possesses an extensive range of biological activities, such as antimicrobial, antitumor, and insecticidal properties [4]. However, as a toxic secondary metabolite, BEA also has cytotoxic, genotoxic, and immunotoxic properties, and can induce oxidative stress via generation of reactive oxygen species (ROS) and lipid peroxidation (LPO) [5]. One of the molecular mechanisms of BEA toxicity has been demonstrated to be related to its ionophoric property, which increases permeability of the plasma membrane for ions and disturbs the normal physiological concentrations [6]. However, the specific mechanism underlying BEA toxicity has not been fully understood.

The frequent occurrence of BEA in maize has highlighted the need for more investigations regarding the potential acute or chronic effects on both humans and animals [7]. Domestic animals are likely to eat moldy feed, and poisonous mycotoxin metabolites ultimately enter the human body, posing great threat to human health. However, BEA has not yet been legislatively regulated in feed and food due to lack of sufficient existing toxicity data, which emphasizes the need for further intensive studies. The liver is the main detoxification and metabolic apparatus. In addition, BEA is reported to exhibit highest hepatotoxicity among eight mycotoxins (penicillic acid, aflatoxin B1, deoxynivalenol, altenariol, andrastin A, citrinin, enniatin B, beauvericin) with clear signs of cellular stress at concentration of 0.9–9 μmol/L [8] and highest accumulation in the liver [9]. Therefore, rat liver cells (BRL3A cells) were chosen as the experimental model in our study.

Oxidative stress is associated with BEA toxicity, which is ascribed to the accumulation of ROS and redox imbalance, leading to cell death via apoptosis, necrosis, or autophagy and resulting in various pathological changes [10]. Under stressful conditions, intrinsic antioxidative defense mechanisms are insufficient to combat excessive ROS attack [11]. Nuclear factor erythroid 2-related factor 2 (NRF2) is a critical transcription factor and a chief regulator of cytoprotective genes, and forms the major cellular defense mechanism against electrophilic stress and oxidative damage by neutralizing ROS to restore redox balance [12]. Usually, Nrf2 is rapidly degraded through the ubiquitin proteasome system in the cytoplasm, in which Kelch-like ECH-associated protein 1 (Keap1) serves as a negative regulator [13]. The cysteine residues of Nrf2 are modified by electrophiles, ROS and nitric oxide, resulting in the dissociation of Nrf2 from Keap 1 and translocation to nucleus to control the antioxidant response by activating the transcription of a cluster of genes bearing antioxidant response element (ARE) [14]. Recent studies have reported a non-canonical pathway of Nrf2 activation, wherein p62 competitively binds to Keap 1 and subsequently releases Nrf2 to translocate into the nucleus [15], which connects autophagy with the Nrf2 signaling pathway. Additionally, Komatsu et al. [16] showed that persistent activation of Nrf2 in autophagy-deficient livers led to severe cell injury. To this end, whether the impact of Nrf2 is positive or negative is disputable.

Autophagy plays a crucial role in eliminating and degrading damaged mitochondria and oxidized proteins, wherein ROS functions as an inducing signal [17]. A growing body of evidence illustrated that excessive accumulation of ROS initiated autophagy by inhibiting PI3K-Akt-mTOR signaling pathway [18]. p62 is a scaffold protein induced by Nrf2 in response to oxidative stress and seems to link autophagy and the Nrf2 signaling pathway [19]. Xiao et al. [20] reported that Mito Q, a mitochondria-targeted antioxidant, partially reversed the overexpression of ROS and restored mitochondrial function through activating autophagy and suppressing the interaction between Nrf2 and Keap1 in db/db mice. Surprisingly, a study conducted by Dong et al. [21] showed that although cadmium upregulated the expression of antioxidative stress genes significantly, Nrf2 nucleus translocation in a p62-dependent manner was one of the causes in toxin-induced renal damage through AMPK-Akt-mTOR pathway. Numerous studies have reported both positive and negative effects of Nrf2. Herein, we studied for the first time how BEA-induced oxidative stress and ROS-mediated Nrf2 activation and translocation to nucleus were involved in regulating hepatotoxicity in BRL3A cells, which will be valuable for improving our understanding of BEA action mode.

## 2. Results

### 2.1. Liver Cell Injury and Apoptosis Induced by BEA

Cell viability and morphological changes have been proven to reflect the cytotoxicity of different toxins to various cells. Cell viability decreased gradually after exposure to 0.5–4 μmol/L BEA for 6, 12, and 24 h (Appendix A). Lactic Dehydrogenase (LDH) release increased significantly when the BEA concentration was equal or greater than 1 μmol/L at different times, indicating severe damage to the cell membrane. Based on the above results, 0, 1, 1.5, and 2 μmol/L BEA for 12 h was chosen as the optimal dosage in the follow-up experiments. The morphological changes in BRL3A cells after treatment with different concentrations of BEA for 12 h were observed using an inverted microscope. In contrast with the blank control group, the bright-field images showed that an increasing number of rounded, healthy cells incubated with BEA became shrunken and diminished in cell size, which indicates that BEA could alter cell morphology.

The effect of BEA on apoptosis in BRL3A cells was also evaluated in our study. We observed that the expression of apoptosis-related proteins, cleaved-caspase 3, and cleaved-caspase 9 were modestly elevated, and cleaved-caspase 9 expression was increased significantly (*p* < 0.05) at the peak concentration (2 μmol/L BEA) (Figure 1B). Meanwhile, the ratio of Bcl-2/Bax dramatically decreased at 1.5 and 2 μmol/L BEA, which indicated severe cell death induced by the toxin. To further confirm our observation, flow cytometry was performed. As shown in Figure 1D, a sharp increase was seen in the number of apoptotic cells after 12 h incubation with 1.5 and 2 μmol/L BEA. These data confirm that BEA exposure at high concentrations induces apoptosis and cell death.

### 2.2. Effect of BEA on the Oxidative Damage in BRL3A Cells

Generation of ROS was firstly observed by fluorescence microscopy to detect whether BEA can result in oxidative stress. Figure 2A shows an increase in fluorescence intensity, which demonstrates increased production and accumulation of ROS. To further verify the data, the level of intracellular ROS was evaluated using flow cytometry. As shown in Figure 2B, relative content of ROS was sharply upregulated in a dose-dependent manner. Additionally, production of superoxide anions was measured using fluorescence microscopy. The concentration of superoxide anions in hepatocytes increased upon exposure to BEA at multiple concentrations (1, 1.5, and 2 μmol/L), as indicated by the enhanced red fluorescence (Figure 2C). To confirm whether the expression of antioxidative-related substances was altered after BEA exposure, the production of malondialdehyde (MDA), reduced glutathione (GSH) concentration, and changes in the activities of antioxidant enzymes were evaluated by the corresponding assay kits. The results displayed that low concentration of BEA (1 μmol/L) significantly increased GSH concentration and catalase (CAT) and total superoxide dismutase (T-SOD) activities (*p* < 0.05 or *p* < 0.01). However, as BEA concentration increased, the above indicators displaued an opposite trend. Overall, glutathione peroxidase (GSH-Px) activity and MDA production were elevated all the time in a concentration-dependent manner, which suggested a disturbance in antioxidative capacity and that GSH may be the key target for alleviation of cell injury.

### 2.3. Effect of BEA on the Nrf2 Signaling Pathway of BRL3A Cells

Nrf2, a nuclear transcription factor, plays an important role in regulating numerous protective proteins. We observed that the expression of Nrf2 within the nucleus gradually rose upon BEA exposure, suggesting the activation of Nrf2. Glutathione S-transferase (GST), Homox1 (HO-1), and NAD(P)H dehydrogenase [quinone] 1 (NQO1) are typical downstream target genes of the Nrf2 signaling pathway. As shown in Figure 3B, the expression of GST and HO-1 proteins increased sharply after BRL3A cells were exposed to 1.5 and 2 μmol/L BEA. The expression of Keap 1, an important regulator of oxidative stress and a sensor of redox reaction, was decreased at all concentrations of BEA, indicating the degradation of Keap 1 and separation of Nrf2 from Keap 1. The mRNA expression levels of a series of antioxidative genes downstream of the Nrf2 signaling pathway (HO-1, GST, NQO1, and SOD) were assessed by qRT-PCR, as these genes are commonly used to evaluate Nrf2 activation. Figure 3C shows altered expression of all four genes to varying degrees. As observed using laser confocal microscopy, most Nrf2 protein translocated into the cell nucleus after BEA exposure, confirming that Nrf2 was indeed activated and widely distributed among the nucleus (Figure 3D). Although the gene expression levels of some antioxidative-enzymes increased significantly, the accumulation of Nrf2 did not ameliorate the cell injury caused by BEA (Appendix A).

### 2.4. Effect of BEA on the Autophagy in BRL3A Cells

ROS production and the consequent oxidative damage has been demonstrated to be a regulator in inducing autophagy and macro-autophagy; it can, in turn, relieve the injury caused by oxidative stress via different signaling pathways. Our data revealed an increase in the mRNA and protein expression levels of autophagy-related proteins, Beclin1, ATG5, and LC3, after treatment with 1.5 and 2 μmol/L BEA for 12 h (Figure 4). Meanwhile, ATG5 protein expression also increased significantly (*p* < 0.05). Figure 4D showed the number of LC3 puncta increased in the cytoplasm of BRL3A cells treated with increasing BEA concentrations. The above results suggested the activation of autophagy and degradation inhibition of autophagy products.

### 2.5. Effect of ML385 on the Alleviation of Cell Injury in BEA-Exposed BRL3A Cells

The role of nuclear translocation of Nrf2 in BEA-induced cell damage remains controversial. Accordingly, ML385, a compound that lowers the activity of Nrf2 transcription, was used to investigate this. Our results revealed that ML385 significantly reversed BEA-induced increase in the protein expression levels of HO-1, GST, NQO1, and nuclear Nrf2 (Figure 5B); these results were in agreement with the immunofluorescence (Figure 5C) and qRT-PCR results. As shown in Figure 5D, expression level of antioxidative genes decreased with or without significance in the ML385 (5 μmol/L) and BEA (1.5 μmol/L) combined treatment group compared with the BEA group, suggesting reduced activation of antioxidative enzymes downstream of Nrf2. The effect of ML385 on BEA-induced autophagy was also investigated. Figure 6B shows that BEA-induced increase in the expression of LC3, Beclin 1, p62, and ATG5 was decreased upon ML385 treatment to varying degrees, which was accompanied by reduced mRNA levels of autophagy-related genes (Figure 6D), suggesting inhibition of BEA-induced autophagy. The immunofluorescent images shown in Figure 6C demonstrates that ML385 could decrease the number of LC3 puncta in BEA-treated cells, indicating an inhibiting effect on BEA-induced autophagy.

To further validate the role of Nrf2, cell viability was detected using CCK-8 solution, and it was observed that ML385 reversed BEA-induced cell injury (Figure 6E). CDDO-ME, a Nrf2 agonist, was employed to further study the impact of Nrf2. Figure 6F shows that in contrast to BEA group, the cell viability decreased drastically in the CDDO-ME (4 μmol/L) and BEA (1.5 μmol/L) co-treatment group. These results confirm that Nrf2 activation negatively affects cell survival after BEA exposure and ML385 mitigates hepatocyte death.

## 3. Discussion

*Fusarium* mycotoxins are fungal pollutants found in crops, fruits, vegetables, and feed around the world, posing serious threat to human and animal health [22]. BEA, a non-legislated emergent mycotoxin, contaminates feed and food globally [23], as proven by in vitro and in vivo toxicity reports. In our study, we investigated for the first time the role of Nrf2 triggered by oxidative damage on BEA-induced hepatotoxicity in BRL3A cells. A recent review had summarized in vitro mechanistic toxicology of BEA, emphasizing its effects on cell viability and proliferation [5]. Accordingly, cell viability and morphological changes of BEA-exposed BRL3A cells were investigated to evaluate BEA-induced cytotoxicity. We observed strong cytotoxicity after 6 h of treatment with 1.5 μmol/L BEA as evidenced by irregularly shaped cells and a decreased cell density compared with the blank control group, with 50% cell death occurring at around 4 μmol/L BEA. Upon cellular damage, the plasma membrane is impaired and LDH is released into the culture media; the extent of LDH release is a commonly used parameter for quantifying the cytotoxicity of chemicals [4]. To validate BEA toxicity, effect of BEA on apoptosis of BRL3A cells was investigated for the reason that excessive cell apoptosis is the primary event in acute hepatocyte death [24]. Our results revealed an upregulation of expression of apoptosis-related proteins such as caspase 3, a central regulator of apoptosis, caspase 9, and a downregulation of Bcl-2/Bax by BEA, confirming that BEA triggers cell apoptosis.

Accumulation of ROS leads to oxidative stress and altered redox homeostasis, which is detrimental to various vital cellular components [25]. Our results clearly showed that the concentration of ROS and superoxide anions increased as the concentration of BEA increased. The level of MDA, an ROS-induced peroxidation product produced by oxidative damage under conditions of cellular stress [26], was drastically increased upon BEA exposure in our study, indicating reduced antioxidant capacity and higher injury degree of hepatocytes. With regard to the antioxidant system, the alterations in the activity of antioxidative enzymes and content of non-enzymatic antioxidants were partially different from the results reported in previous literature [7,27]. We speculate that in the case of low BEA concentration (1 μmol/L BEA), BRL3A cells partly overcame BEA-induced damage by strengthening the antioxidant power and they exhibited a defensive reaction, while at high concentrations of BEA (1.5 and 2 μmol/L), hepatocytes succumbed to BEA toxicity. Interestingly, Dornetshuber et al. [28] reported that ROS generation was not involved in BEA-induced cytotoxicity. This contradiction could be due to differences in incubation time, exposure concentration, experimental model, and action mode.

Nrf2 is a major regulator of phase Ⅱ antioxidative enzymes and protects cells by counteracting oxidative stress via its accumulation in the cytoplasm, translocation to nucleus, and binding to ARE with subsequent transcriptional activation of target genes [29]. Nrf2 has a short half-life of approximately 20 min and is negatively regulated by Keap 1 [30]. Studies have reported that Nrf2-induced increase in the activity of antioxidant enzymes is responsible for the amelioration of BEA-induced cell injury [31]. However, recent studies have reported that Nrf2-mediated antioxidative effects might instead aberrantly aggravate cell injury [32]. It is, therefore, debatable whether Nrf2 exerts a positive or a negative effect, and further studies are necessary to elucidate the regulation or dysregulation of the Nrf2 signaling pathway under certain circumstances. Autophagy is another intracellular defense mechanism, mediated by the Nrf2 pathway via p62, a specific autophagy substrate [33]. The crosstalk between toxin-induced autophagy and prolonged Nrf2 activation was investigated by Fan et al. [34], revealing a vicious cycle of repressive autophagy and oxidative damage. Hence, it is necessary to unravel the Nrf2-autophagy axis. In our study, the expression levels of the Nrf2 signaling pathway-related proteins increased drastically and so did the mRNA level of Nrf2-targeted antioxidative genes, which was accompanied by Nrf2 nuclear translocation, indicating the accumulation of Nrf2, promotion of Nrf2 translocation into the nucleus, and initiation of antioxidant response. Moreover, BEA-induced upregulation of autophagy-related proteins (LC3, ATG5, Beclin 1, and p62) and of autophagy-related genes indicate increased production of autophagosome and checked degradation of autophagy products, which was further verified by increasing number of LC3 puncta. However, activation of intracellular defense mechanisms did not alleviate hepatocyte damage incurred by BEA as evidenced by the low cell vitality.

To further confirm the role of Nrf2 nuclear translocation, ML385, an antagonist that inhibits Nrf2 transcription, was administrated. Notably, the decrease in cell viability after BEA exposure was partially reversed upon ML385 treatment in our study. This is contrary to a previous study reporting the adverse effect of ML385 in AML12 cells and mouse normal hepatic cells [35]. Though both were hepatocytes, the difference in results could be due to diverse experiment conditions, especially considering that the damage factor, used in our study, BEA, is a special noxious mycotoxin with many beneficial activities. CDDO-ME, a stimulator of Nrf2, greatly lowered cell viability and enhanced hepatotoxicity of BEA with increased cell death rate. However, various studies reported the positive effect of CDDO on cell survival rate, thereby protecting tissues and cells against toxin-induced cell damage [36]. These unusual results prompted us to further investigate the role of Nrf2 nuclear translocation in BEA-induced hepatocyte injury. Overall, our study clearly shows BEA leads to BRL3A cell apoptosis and oxidative stress through ROS overproduction and strongly emphasizes the role of Nrf2 nuclear translocation in BEA-induced cell damage, which could be aggravated or alleviated by Nrf2 excessive activation or Nrf2 inhibition, providing a theoretical basis for further investigation of the in vitro molecular mechanism of BEA hepatotoxicity. However, the relationship between BEA-induced Nrf2 activation and p62-mediated autophagy is not yet clear.

## 4. Conclusions

Our study substantiates that BEA decreases cell viability, alters cell morphology, and induces apoptosis in BRL3A cells. Additionally, BEA triggers oxidative stress through overproduction of ROS and superoxide anions and by disrupting redox equilibrium. Consequently, the Nrf2 signaling pathway was activated, accompanied by translocation of Nrf2 from cytoplasm to the nucleus, transcriptional activation of downstream antioxidative genes, and activation of autophagy, which negatively regulates cell survival. ML385 reversed the negative effects of Nrf2 and partially reduced BEA-induced cellular damage. However, the connection of Nrf2 activation and autophagy in BEA-induced hepatotoxicity remains unclear and warrants further investigation.

## 5. Materials and Methods

### 5.1. Chemicals and Reagents

BEA (MW: 783.96 g/mol, CAS No.: 26048-05-5) was purchased from Pribolab Biological Engineering Co. Ltd (Pribolab, Qingdao, China). ML385 (MW: 511.59 g/mol, CAS No.: 846557-71-9), and CDDO (MW: 491.66 g/mol, CAS No.: 218600-44-3) were obtained from MedChemexpress (St. Louis, MO, USA). DMSO, the solvent of ML385 and CDDO, was purchased from Solarbio, Beijing Science & Technology Co, Ltd (Solarbio, Beijing, China). Acetonitrile was from Sinopharm Chemical Reagent Co, Ltd (Shanghai, China). Dulbecco’s Modified Eagle’s Medium (DMEM) was obtained from GIBCO BRL (Gaithersburg, MD, USA); 4, 6-diamidino-2-phenylindole (DAPI) was obtained from Beyotime Institute of Biotechnology (Shanghai, China). Annexin V-FITC/PI and cell counting kit-8 (CCK-8) solution were purchased from Vazyme Biotech Co. Ltd (Nanjing, China). DHE, LDH cytotoxicity assay kit and ROS assay kit were procured from Beyotime (Shanghai, China). Rabbit monoclonal antibodies against Nrf2, HO-1, NQO1, p62, Beclin1, and Histone H3, and mouse monoclonal antibodies against β-actin, GST were purchased from Cell Signaling Technology (Beverly, MA, USA). Rabbit monoclonal antibodies against Bax, Bcl-2, and ATG5 antibody were obtained from Abcam Crop. (Cambridge, MA, USA). Keap 1 and Beclin 1 were purchased from Santa Cruz Biotechnology (Dallas, TX, USA). LC3, NQO1, and p62 were from ABclonal Technology (ABclonal, Wuhan, China). Cleaved-caspase 3 and cleaved-caspase 9, GAPDH and β-actin were purchased from Proteintech Group, Inc (Proteintech, Wuhan, China). Peroxidase conjugated secondary antibody was purchased from Jackson ImmunoResearch Inc. (Langcaster, PA, USA). All chemicals and reagents were of analytical grade.

### 5.2. Cell Culture

BRL3A cells (rat liver cell line) were purchased from the American Type Culture Collection (ATCC, Rockefeller, Maryland, MD, USA). The cells were cultivated in DMEM with 10% fetal bovine serum (FBS), 1% penicillin-streptomycin, and 1 mmol/L glutamine, and grown at 37 °C in the presence of 5% CO_2_.

### 5.3. Cell Proliferation Assay

BRL3A cells were plated at a density of 5 × 10^3^ cells per well in a 96-well plate. When the cells reached 60–70%, the experiment was divided into 8 groups: blank control group, acetonitrile group (BEA solvent, 0.1%), and 0.5, 1, 1.5, 2, 3, and 4 μmol/L BEA groups separately at the former stage. At the latter stage, BRL3A cells were treated with 5 μmol/L ML385 or 4 μmol/L CDDO, combined with 1.5 μmol/L BEA for 12 h. DMSO was the solvent (0.1%) of ML385 and CDDO.

CCK-8 solution (10 nM) was added to each well, and then the culture plate was placed in an incubator for 1 h. An EL×800 Absorbance Microplate Reader (BioTek Instruments, Inc, Santa Clara, CA, USA) was used to measure the absorbance at 450 nm. In the follow-up experiment, 0.1% acetonitrile was added to the blank control group (all BEA groups already had 0.1% acetonitrile). The morphology of BRL3A cells was observed using an optical microscope (Leica, Wetzlar, Germany).

### 5.4. LDH Release Assay

BRL3A cells were exposed to various concentrations of BEA and incubated for 6, 12, and 24 h. A microplate reader was used to measure the release of LDH as an indicator of cytotoxicity according to the LDH cytotoxicity detection kit. The absorbance was set at 490 nm. According to results obtained by CCK8 and LDH release assay, 0, 1, 1.5, and 2 μmol/L BEA were chosen in our study.

### 5.5. Measurement of Apoptosis Rate by Flow Cytometry

BEA-treated BRL3A cells were collected by centrifuging at 150 RCF for 5 min after trypsinization and washed twice with phosphate buffer solution (PBS). The cells were then labeled with annexin V-FITC and PI according to the manufacturer’s instruction. The apoptosis rate of BRL3A cells treated with BEA was detected by flow cytometry (Becton, Dickinson, San Jose, CA, USA).

### 5.6. Detection of Superoxide Anions

DHE is a commonly used fluorescent probe for the measurement of superoxide as an indicator of oxidative damage. After 12 h of BEA exposure, the cells were washed with PBS and incubated with DHE dye at 37 °C for 30 min in the dark. The samples were then observed using fluorescence microscopy (TCSSP8STED, Leica, Wetzlar, Germany).

### 5.7. Measurement of ROS

The production of intracellular ROS was measured using an ROS assay kit (Beyotime, Shanghai, China) by flow cytometry. Following digestion with 0.25% trypsin, the cells were washed with pre-cooled PBS, resuspended in serum-free medium and loaded with 10 μmol/L DCFH-DA at 37 °C for 30 min in the dark. Then, the level of ROS was detected by flow cytometry. FlowJo version 10 (Becton, Dickinson and Company, Franklin Lakes, NJ, USA) was used for the analysis of the results. Additionally, the samples loaded with the fluorescence probe were observed using fluorescence microscopy.

### 5.8. Measurement of Oxidants and Antioxidants

T-SOD, GSH-Px, CAT activities, and MDA, and GSH content in BRL3A cells were measured using specific assay kits from Nanjing Jiancheng Bioengineering Institute (Nanjing, China) according to the manufacturer’s protocol.

### 5.9. Western Blot Analysis

BRL3A cells were lysed using radioimmunoprecipitation assay (RIPA) lysis buffer (Beyotime). Nucleoprotein was extracted according to the nuclear and cytoplasmic protein extraction kit’s instruction (Beyotime). Following by the measurement of protein concentrations by a bicinchoninic acid (BCA) protein assay kit (Beyotime), the samples were mixed with sodium dodecyl sulfate—polyacrylamide (SDS-PAGE) loading buffer (5×) under reducing condition. A total of 20–30 μg protein samples were separated by 12% SDS-PAGE gel electrophoresis and subsequently transferred to polyvinylidene fluoride membranes. After blocking with 5% skim milk for 2 h, the membranes were incubated overnight at 4 °C with primary antibodies. After washing with tris-buffered saline tween 20 (TBST), the samples were incubated with the corresponding horseradish peroxidase (HRP)-conjugated secondary antibody for 2 h at 20 °C. Primary antibodies against Bax, Bcl-2, cleaved-caspase 3, cleaved-caspase 9, as well as Nrf2, Keap 1, NQO1, HO-1, GST, p62, Beclin 1, ATG5, LC3, Histone H3, GAPDH, and β-actin were used. Peroxidase conjugated secondary antibody (Jackson ImmunoResearch Inc., Langcaster, PA, USA) was used (Appendix A). After washing again, the membranes were exposed to an enhanced chemiluminescence (ECL) reagent (New Cell & Molecular Biotech Co., Ltd., Suzhou, China) and the results were analyzed using ImageJ (National Institute of Health, Bethesda, MD, USA).

### 5.10. RNA Isolation, cDNA Synthesis and Quantitative Real-Time PCR (qRT-PCR)

The primers for qRT-PCR (Appendix A) were designed by Oligo and synthesized by Sangon Biotech (Shanghai, China). Total RNA was extracted using TRIzol (Solarbio, Beijing Science & Technology Co. Ltd., Beijing, China) and its purity was measured by NanDrop 2000 Spectrophotometer (ThermoFisher Scientific, Waltham, MA, USA). Then, a reverse transcription kit of Strand cDNA Synthesis SuperMix for qPCR (Yeasen Biotech Co. Ltd., Shanghai, China) was used to synthesize cDNA from RNA. Using qPCR SYBR Green Master Mix (Yeasen, Shanghai, China), qRT-PCR was performed using a 7500 Real-Time PCR System (Bio-Rad, Hercules, CA, USA). PCR amplification program was shown in Appendix A. The fold change of target gene expression was analyzed according to the formula proposed by Pfaffl: 2^−ΔΔCt^.

### 5.11. Immunofluorescence Analysis

The cells were cultured in a 24-well plate at a density of 5 × 10^4^ cells per well. After treating with BEA for 12 h, the cells were fixed in 4% paraformaldehyde at 4 °C for 30 min followed by washing with PBS. The samples were incubated with 0.5% Triton X-100 at room temperature for 10 min and treated with 5% bovine serum albumin (BSA) sealing solution for 1 h, then incubated with anti-Nrf2 antibody (1:500) or anti-LC3B antibody (1:200) at 4 °C overnight. The cells were washed three times prior to incubation with fluorescent-labelled secondary antibody (Abmart Shanghai Co., Ltd, Shanghai, China) for 1 h in the dark, then stained with DAPI for 10 min. Confocal fluorescence microscope (TCSSP8STED, Leica, Wetzlar, Germany) was used to detect the localization of Nrf2 protein and LC3 puncta in BRL3A cells.

### 5.12. Statistical Analysis

All experiments were performed in triplicate using independent repeats. The results were analyzed by one-way ANOVA using SPSS 22.0 and presented as mean ± SD. *p* < 0.05 was set as a significant threshold.

## Figures and Tables

**Figure 1 toxins-14-00367-f001:**
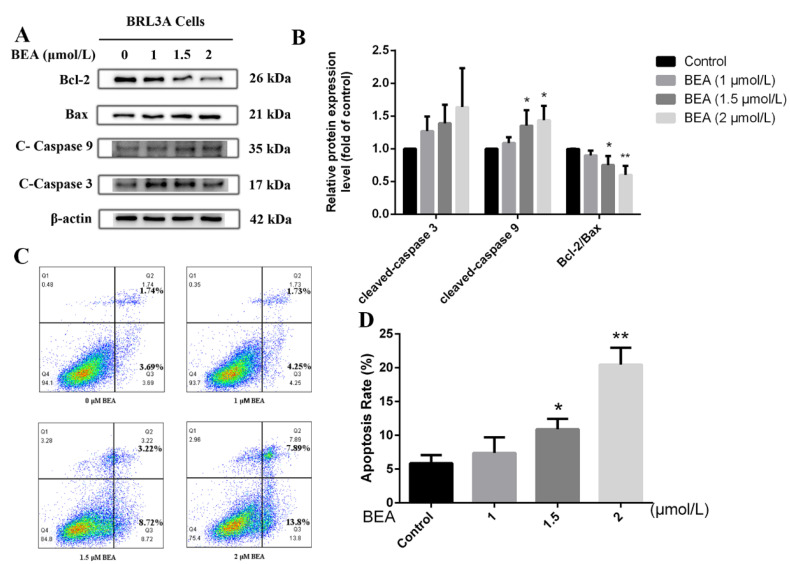
Effect of BEA exposure on apoptosis in BRL3A cells in a concentration-dependent manner. (**A**) Expressions of apoptosis-related proteins were detected by western blot after BEA treatment at the concentrations of 0, 1, 1.5, and 2 μmol/L for 12 h. (**B**) The ratio of Bcl-2/Bax, cleaved-caspase 3/β-actin, and cleaved-caspase 9/β-actin was analyzed by ImageJ, respectively. All western blot results were normalized with respect to β-actin. (**C**,**D**) Apoptosis rate of BRL3A cells after BEA exposure was determined using flow cytometry with annexin V-FITC/PI staining. All experiments were performed in triplicate (*n* = 3) and data were presented as the means ± SD and analyzed by one-way ANOVA. * *p* < 0.05 and ** *p* < 0.01, compared with the blank control group.

**Figure 2 toxins-14-00367-f002:**
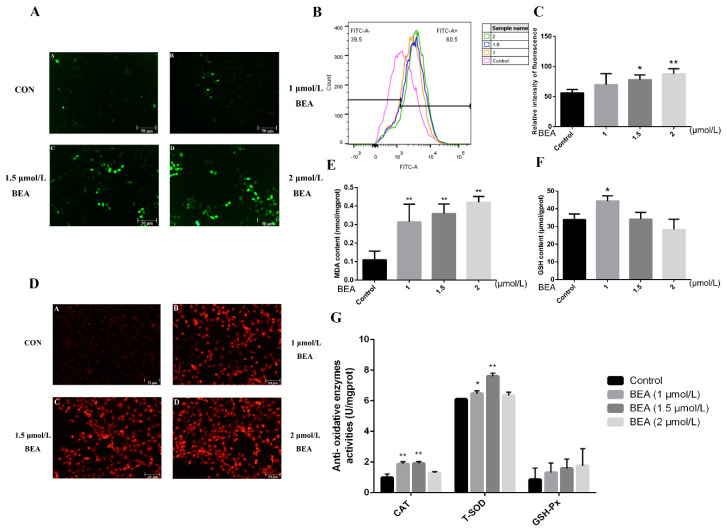
Effect of BEA on oxidative damage in BRL3A cells. (**A**) After the cells were exposed to 0, 1, 1.5, and 2 μmol/L BEA for 12 h, the samples were stained using a DCFH-DA fluorescent probe and the distribution of ROS was observed using a fluorescence microscope. Scale bar = 50 μm. (**B**,**C**) Flow cytometry was employed to measure the exact level of intracellular ROS induced by BEA. * *p* < 0.05 and ** *p* < 0.01, compared with the blank control group. (**D**) The relative concentration of superoxide anions attributed by BEA was measured by a dihydroethidium (DHE) probe using fluorescent microscopy. Scale bar = 50 μm. (**E**–**G**) The effects of BEA on lipid peroxidation (MDA), concentration of non-enzymatic antioxidant (GSH), and activities of antioxidative enzymes (CAT, T-SOD and GSH-Px) were detected by corresponding commercial assay kits. All experiments were performed in triplicate (*n* = 3) and data were presented as the means ± SD and analyzed by one-way ANOVA. * *p* < 0.05 and ** *p* < 0.01, compared with the blank control group.

**Figure 3 toxins-14-00367-f003:**
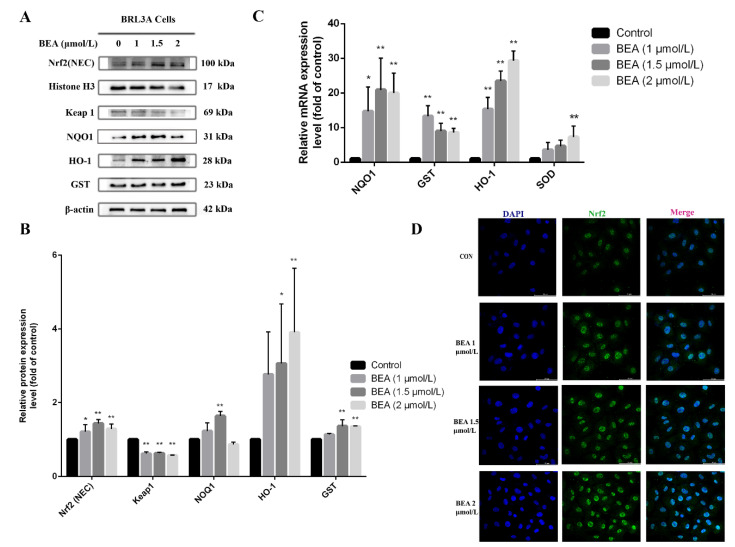
Effect of BEA on the Nrf2/Keap 1 signaling pathway in BRL3A cells. (**A**) Cells were treated with increasing concentrations of BEA (0, 1, 1.5 and 2 μmol/L) for 12 h. The expression of Nrf2/Keap-1 signaling pathway-related proteins was detected by western blot. (**B**) The ratio of nuclear Nrf2/Histone H3 and expression levels of GST, HO-1, Keap 1, and NOQ1 in total protein extract were analyzed by ImageJ. (**C**) mRNA expression levels of antioxidant-related genes (GST, NQO1, SOD, and HO-1) downstream of Nrf2 signaling pathway were measured by qRT-PCR. All experiments were performed in triplicate (*n* = 3) and data were presented as the means ± SD and analyzed by one-way ANOVA. * *p* < 0.05 and ** *p* < 0.01, compared with the blank control group. (**D**) Nuclear translocation of Nrf2 protein in BRL3A cells was photographed by laser confocal fluorescence microscopy. BRL3A cell nucleus (blue), Nrf2 (green), scale bar = 50 μm.

**Figure 4 toxins-14-00367-f004:**
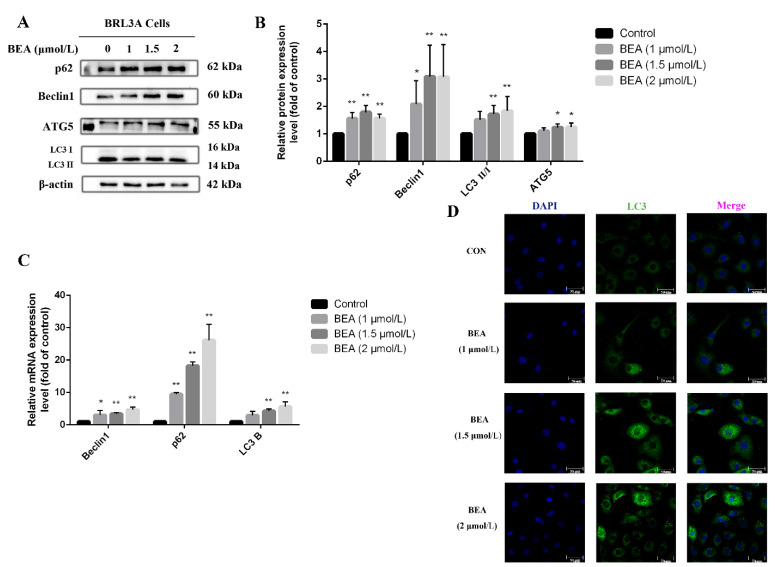
Effect of BEA on the autophagy in BRL3A cells. (**A**) Western blot analysis was performed to detect the expression level of autophagy-related proteins after BRL3A cells were treated with 0, 1, 1.5, and 2 μmol/L BEA for 12 h. (**B**) The ratios of p62/β-actin, Beclin 1/β-actin, ATG5/β-actin, and LC3 Ⅱ/Ⅰ. (**C**) mRNA expression of autophagy-related genes was measured by qRT-PCR assay. All experiments were performed in triplicate (*n* = 3) and data were presented as the means ± SD and analyzed by one-way ANOVA. * *p* < 0.05 and ** *p* < 0.01, compared with the blank control group. (**D**) Number of LC3 puncta in cytoplasm of BRL3A cells was photographed by laser confocal fluorescence microscopy. BRL3A cell nucleus (blue), LC3 puncta (green), scale bar = 50 μm.

**Figure 5 toxins-14-00367-f005:**
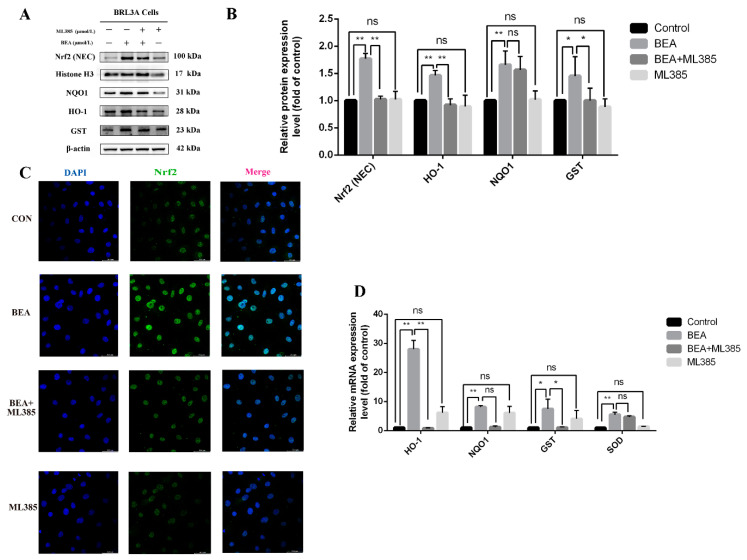
Effect of ML385 on reduced Nrf2 activation in BEA-exposed BRL3A cells. (**A**) The levels of Nrf2 signaling pathway-related proteins followed by treatment with 0, BEA (1.5 μmol/L), or ML385 (5 μmol/L), or BEA + ML385 for 12 h was detected using western blot. (**B**) ImageJ analysis of Nrf2 (nuclear), NQO1, HO-1, and GST. (**C**) Translocation of Nrf2 to the nucleus in BRL3A cells was observed using confocal fluorescence microscopy. BRL3A cell nucleus (blue), Nrf2 (green), scale bar = 50 μm. (**D**) qRT-PCR was performed to measure mRNA expression levels of GST, HO-1, NQO1, and SOD. All experiments were performed in triplicate (*n* = 3) and data were presented as the means ± SD and analyzed by one-way ANOVA. * *p* < 0.05; ** *p* < 0.01; ns, not significant.

**Figure 6 toxins-14-00367-f006:**
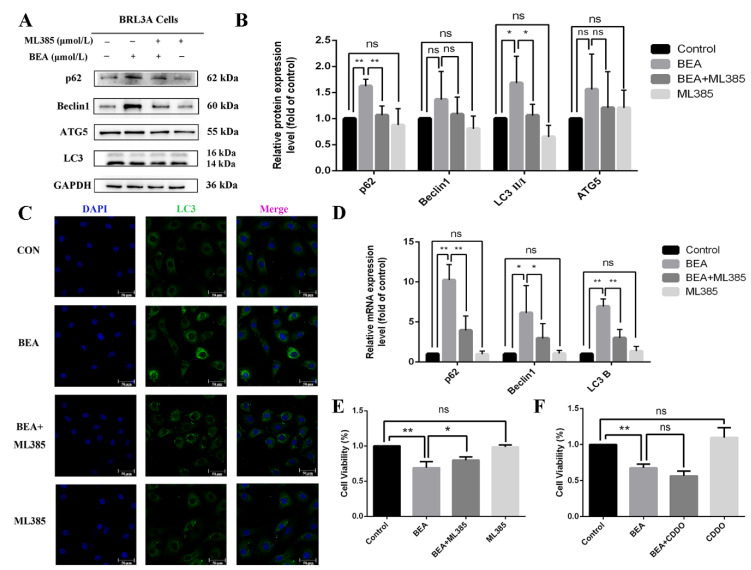
Effect of ML385 on the alleviation of cell injury in BEA-induced BRL3A cells. (**A**) Expression levels of autophagy-related proteins in BRL3A cells were detected after treatment with 0, BEA (1.5 μmol/L), or ML385 (5 μmol/L), or BEA + ML385 for 12 h. (**B**) Western blot analysis for expression of p62, Beclin1, ATG5, and LC3 II/I. * *p* < 0.05; ** *p* < 0.01; ns, not significant. (**C**) Number of LC3 puncta in cytoplasm of BRL3A cells was photographed by laser confocal fluorescence microscopy. BRL3A cell nucleus (blue), LC3 puncta (green), scale bar = 50 μm. (**D**) qRT-PCR was performed in the measurement of mRNA expression levels in autophagy-related genes. The effect of 5 μmol/L ML385 (**E**) or 4 μmol/L CDDO (**F**) on alleviation or aggravation of BEA-induced reduced cell viability. All experiments were performed in triplicate (*n* = 3) and data were presented as the means ± SD and analyzed by one-way ANOVA. * *p* < 0.05; ** *p* < 0.01; ns, not significant.

## Data Availability

Not applicable.

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
