# Peer review of "Role of Nrf2 Nucleus Translocation in Beauvericin-Induced Cell Damage in Rat Hepatocytes"

_toxins, 2022, doi:10.3390/toxins14060367_

Round 1

Reviewer 1 Report

This is a very good designed and very well presented study. Introduction gives appropriate information about the subject of the study, however there is a need to underline the aim of the study. The obtained results are presented clearly with sufficient statistical analysis. The discussion is profound and accurate. In conclusions, I would avoid the beginning of the first sentence with expression “in summary”. Besides, it is a rare situation where someone refers in conclusions to figure, even if it's a some kind of graphical abstract of your study this should be described in the results or discussion chapter.

Line 5: Genus names should be capitalized

Reviewer 2 Report

The manuscript “Role of Nrf2 nucleus translocation in Beauvericin-induced cell damage in rat hepatocytes”The study is good and the findings are satisfactory. However, a few points need to address before the acceptance.

  1. The author showed antioxidant, autophagy, and apoptotic roles of Beauvericin, however, there is no correlation established between these. Therefore, the author must establish the connection between these.  
  2. For the nuclear fraction (Nrf2) represented in figure 3 and figure 5 β-actine is not the appropriate loading control. Therefore, add a suitable loading control such as Lamine B or any other.
  3. It would be better if the author can represent the Nrf2 in both cytosol and nuclear fraction. That will give clear evidence for the translocation of Nrf2 from the cytosol to nuclear in response to Beauvericin.
  4. Correct CCK-8 method (line no 8). CCK-8 is not a method.
  5. Rewrite lines no 8-10, Rewrite the sentence “Our results ……. antioxidative genes (lines 10-14). The sentence is too long and not clear.
  6. Keywords should have opted more wisely. The words that appear in the title should be avoided in the keywords.
  7. Line 44, “eight mycotoixns”???? Specify which mycotoxins.
  8. LDH (Lactic Dehydrogenase) rewritten as LDH (Lactate Dehydrogenase) (line 86).
  9. Morphological images (figure S1) are not clear, the author should take the high magnification and resolution images.
  10. Merge the sub-sections 5.3.1 and 5.3.2 with section 5.3.
  11. Address the typo mistake throughout the manuscript e.g “for w 1 h” (Line 361)
  12. Write the full name of the abbreviation while appearing first in the text e.g. DHE (line 379)

Reviewer 3 Report

The manuscript entitled ‘Role of Nrf2 nucleus translocation in Beauvericin-induced cell damage in rat hepatocytes’ is a very interesting study on the mycotoxin Beauvericin that contaminates food and has a variety of variety of biological activities including hepatotoxicity. The introduction is well written, the methodology used is adequate and the results are convincing and supported by plethora of experiments. The findings here reported are of great interest and I strongly recommend the publication of the submitted manuscript. However, I have few suggestions for the authors, as reported below.

Materials and methods:

Please improve the paragraph on western blot by adding some information more on SDS-PAGE e.g. which percentage of acrylamide was used? Is SDS-PAGE performed under reducing condition? Which protein amount was loaded? Moreover, change ‘5 × sodium dodecyl sulfate (SDS)-polyacrylamide loading buffer’ by ‘SDS-PAGE Loading Buffer (5x)’;

Figures: Add the statistical test used in the Figure legend and improve the resolutions of images of cells, I refer mostly to the scale bar in the Figure, is not visible or is not clear. Change ‘KDa’ by ‘kDa’;

Figure 7: please add a Figure legend.
